OBSERVATION

# Swapping Metagenomics Preprocessing Pipeline Components Offers Speed and Sensitivity Increases

George Armstrong,a,b Cameron Martino,a,b,c Justin Morris,d,e Behnam Khaleghi,f Jaeyoung Kang,e Jeff DeReus,a,c Qiyun Zhu,g,h Daniel Roush,g,h Daniel McDonald,a Antonio Gonazlez,a Justin P. Shaffer,a Carolina Carpenter,c,i Mehrbod Estaki,a Stephen Wandro,c Sean Eilert,j Ameen Akel,j Justin Eno,j Ken Curewitz,j Austin D. Swafford,c Niema Moshiri,f Tajana Rosing,c,e,f Rob Knighta,f,k

aDepartment of Pediatrics, School of Medicine, University of California, San Diego, California, USA
bBioinformatics and Systems Biology Program, University of California, San Diego, California, USA
cCenter for Microbiome Innovation, Jacobs School of Engineering, University of California San Diego, La Jolla, California, USA
dDepartment of Electrical and Computer Engineering, San Diego State University, San Diego, California, USA
eDepartment of Electrical and Computer Engineering, Jacobs School of Engineering, University of California San Diego, La Jolla, California, USA
fDepartment of Computer Science and Engineering, Jacobs School of Engineering, University of California San Diego, La Jolla, California, USA
gSchool of Life Sciences, Arizona State University, Tempe, Arizona, USA
hBiodesign Center for Fundamental and Applied Microbiomics, Arizona State University, Tempe, Arizona, USA
iScripps Institution of Oceanography, University of California San Diego, La Jolla, California, USA
jMicron Technology, Inc., Folsom, California, USA
kDepartment of Bioengineering, University of California, San Diego, La Jolla, California, USA

George Armstrong and Cameron Martino contributed equally.

**ABSTRACT** Increasing data volumes on high-throughput sequencing instruments such as the NovaSeq 6000 leads to long computational bottlenecks for common metagenomics data preprocessing tasks such as adaptor and primer trimming and host removal. Here, we test whether faster recently developed computational tools (Fastp and Minimap2) can replace widely used choices (Atropos and Bowtie2), obtaining dramatic accelerations with additional sensitivity and minimal loss of specificity for these tasks. Furthermore, the taxonomic tables resulting from downstream processing provide biologically comparable results. However, we demonstrate that for taxonomic assignment, Bowtie2's specificity is still required. We suggest that periodic reevaluation of pipeline components, together with improvements to standardized APIs to chain them together, will greatly enhance the efficiency of common bioinformatics tasks while also facilitating incorporation of further optimized steps running on GPUs, FPGAs, or other architectures. We also note that a detailed exploration of available algorithms and pipeline components is an important step that should be taken before optimization of less efficient algorithms on advanced or nonstandard hardware.

**IMPORTANCE** In shotgun metagenomics studies that seek to relate changes in microbial DNA across samples, processing the data on a computer often takes longer than obtaining the data from the sequencing instrument. Recently developed software packages that perform individual steps in the pipeline of data processing in principle offer speed advantages, but in practice they may contain pitfalls that prevent their use, for example, they may make approximations that introduce unacceptable errors in the data. Here, we show that differences in choices of these components can speed up overall data processing by 5-fold or more on the same hardware while maintaining a high degree of correctness, greatly reducing the time taken to interpret results. This is an important step for using the data in clinical settings, where the time taken to obtain the results may be critical for guiding treatment.

**KEYWORDS** alignment, host filtering, metagenomics

**Ad Hoc Peer Reviewer** Zhong Wang

Address correspondence to Rob Knight, robknight@ucsd.edu.

The authors declare no conflict of interest.

The universal first step in processing metagenomic and metatranscriptomic data is quality filtering and trimming (i.e., removing low-quality reads and removing sequences introduced as technical artifacts such as sequencing adaptors and PCR primers), so that only high-quality data that correspond to nucleic acid sequences in the original samples are retained. For samples derived from humans, or where host DNA dominates over microbial DNA (for example, biopsy specimens, surface swabs from skin or plants, etc.), filtering out sequences that are derived from the host rather than microbes is also important for ethical and/or technical reasons. Increasing data volumes with newer sequencing instrumentation have transformed these steps from minor nuisances to efforts that require major computation, typically involving clusters or cloud computing solutions.

A widely used combination for quality filtering, trimming, and host filtering is Atropos (1) plus Bowtie2 (2), both of which are popular and widely used tools for these tasks. A few of the many examples of publications that have used either tool for these tasks include comparisons of multiple pipelines for nucleic acid extraction (3), analysis of a large Finnish cardiac risk cohort (4), the popular KneadData preprocessing tool (5), and a recent paper examining the metavirome of the mosquito *Aedes aegypti* (6).

As data sets have scaled rapidly, the need for near-real-time processing to support clinical applications, such as choice of antibiotics in sepsis, determination of respiratory symptoms as bacterial or viral (including novel pathogens such as SARS-CoV-2), and choice of anticancer medications, have prompted exploration of hardware acceleration approaches such as GPUs (7), FPGAs (8), and in-memory computing approaches (9) for key analysis steps, including alignment. Driven by weeks- to months-long delays in processing data from large projects, in the DARPA-sponsored JUMP-CRISP project, we sought to benchmark and characterize the slow steps in the popular Atropos plus Bowtie2 pipeline. However, prior to proceeding directly to implementation of this pipeline on an alternative architecture, we sought to determine whether other CPU-based tools might provide sufficient performance improvement and/or provide a better candidate for acceleration.

Here, we explored other combinations of popular methods and found that the combination of Fastp (10) (trimming) and Minimap2 (11) (host-filtering) performed best. We then demonstrated that this faster combination of processing produces outputs that are quantitatively similar to previous conventional methods in both data-driven simulation data and real data derived from a broad set of extraction kits and sample types.

While implementing the host-filtering benchmarks, we discovered a read count limitation with Bowtie2. When used on large sequencing data sets, the reads after $2^{32}$ were not included in Bowtie2's output, prohibiting successful application of host-filtering on full NovaSeq lanes. We subsequently fixed this, and the update is available in Bowtie2 v2.4.2 or later. We used this updated version in our benchmarks.

To evaluate runtime performance, we used the popular CAMI-Sim package (12), one of the important outputs of the CAMI (Critical Assessment of Metagenome Interpretation) project (13), to generate simulated data sets containing known amounts of host genome contamination. The simulated data contained 150-bp reads sampled from 10 microbial and 1 human reference genome (see Table S1 in the supplemental material). Errors were simulated into the reads with ART (14) using Illumina default error profiles. Minimap2 (preset for short reads), Bowtie2 (which allows several preset modes that trade sensitivity for speed), and BWA MEM (15) (no presets, so defaults were used) were run with 12 threads to align the simulated reads to a different human reference (T2T CHM13). Figure 1 documents the reduction in read misclassification (Fig. 1A) and false negatives (Fig. 1B) of host filtering by Minimap2 and BWA MEM over Bowtie2. Minimap2 provides a 1.6- to 8.3-fold improvement in speed of computation on the same data, compared to the most sensitive version of Bowtie2, while offering 10.5- to 44.3-fold improvement over BWA (Fig. 1C). Compared to Bowtie2, Minimap2's runtime performs more favorably with the amount of host contamination, making it suitable for even highly host-contaminated samples, such

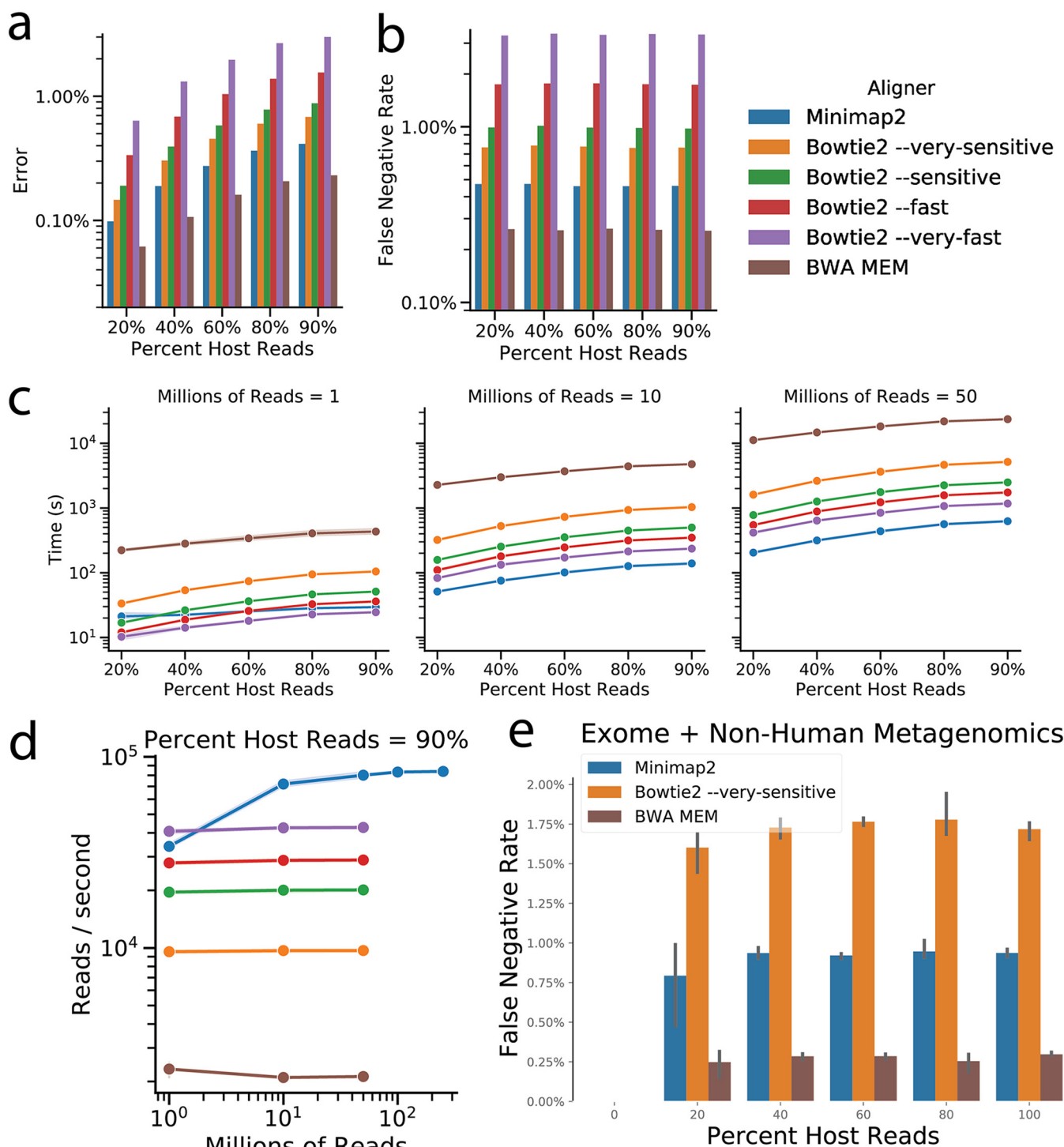

**FIG 1** Minimap2 provides improved error, sensitivity, and runtime for host filtering over the current open-source pipeline. Comparison of aligners for host filtering on 1 million CAMI-Sim simulated reads by error (a) and human reads (b) failed to align to the reference (false-negative rate). (c and d) Time (c) and processing rate (d) comparison across aligners of 1 million, 10 million, and 50 million CAMI-Sim simulated reads. Minimap2 is shown for 100 million and 250 million reads. (e) False-negative rate of host filtering on data with real reads combined from separate exome sequencing and nonhuman metagenomics studies.

as tissue biopsy specimens, saliva, nasal cavity, skin, and vaginal samples, which can contain >90% host DNA (16, 17). It is also notable that while Bowtie2 and BWA MEM process reads at a relatively constant rate across all the tested read counts, Minimap2 does not achieve optimal performance until it operates on a larger number of reads (Fig. 1D). For runtime, we have focused on the host-filtering step because it took the bulk of the time,

and the results of trimming are largely unchanged between Fastp and Atropos (Fig. S1A). When comparing the widely used combination of Atropos plus Bowtie2 to the new fastest approach of Fastp plus Minimap2, we note that the overall pipeline, including trimming and filtering components, was accelerated overall by a factor of 5.6 (Fig. S1B), which may come at the cost of increased memory usage (Fig. S1C).

To further validate the results between Bowtie2, BWA MEM, and Minimap2 on real sequencing data, we created *in silico* mock mixtures of data from known sources. We first obtained IGSR phase 3 (18) human exome sequencing data (Table S2) that are likely free of microbial genomic contamination compared to whole-genome sequencing, which can be contaminated with microbial reads (16, 17). We then obtained soil rhizosphere and mouse fecal metagenomics sequencing data, free of any human genome contamination. From these two data sets we produced benchmarking samples of 1,000, 100,000, and 1 million total sequences with various proportions of microbial versus human-derived sequencing data ranging from 0 to 100% human. The samples were then processed by Bowtie2 (very sensitive), BWA MEM, and Minimap2. As observed in the simulation data, under all conditions Minimap2 and BWA MEM outperformed the most sensitive version of Bowtie2 in allowing fewer human sequences to pass read filtering (Fig. 1E).

Although these results on simulated data were encouraging, it is critical to benchmark new techniques on real-world data. We therefore used one of our recently published data sets comparing different nucleic acid extraction methods, which provided a built-in way of comparing any differences of biological interpretation between the previously established end-to-end pipeline and the new, fastest combination of Fastp and Minimap2. These kit comparison data sets contain samples from a range of biospecimen types with differing host DNA loads (3). Across the three extraction conditions tested in that paper, the total number of reads recovered from each sample was essentially identical between the Atropos/Bowtie2 and the Fastp/Minimap2 pipelines (Fig. 2A), and the alpha diversity estimates within each sample were also essentially identical (Fig. 2B). The sample pairs with different host-filtering methods were also extremely similar in unweighted and weighted ordination results (Fig. 2C), with differences between individual specimens run through both pipelines (connected by lines) typically much smaller than differences between different specimens, even of the same biospecimen types. Finally, the overlap of taxonomic calls at the phylum, genus, and species levels was perfect between the two pipelines (Fig. 2D).

Given the dramatic improvement in preprocessing and host filtering, we further sought to test whether Minimap2 is suitable for taxonomic assignment with similar speed advantages compared to Bowtie2, which is used in the Woltka pipeline (19). Using Woltka benchmarking data sets for taxonomic assignment (Text S1), we found Minimap2 performs comparatively poorly, with a reduced F1 score (Fig. S2A). This is potentially attributed to the higher false-positive rate of Minimap2 (Fig. S2B), since it will result in more alternate alignments between similar genomes, which detract from Woltka's accuracy. Therefore, research into accelerating this part of the overall analysis pipeline for shotgun metagenomics data should focus on accelerating other methods rather than Minimap2.

Taken together, our results suggest several important principles for optimization of shotgun metagenomics workflows. First, even widely used pipeline components should be periodically reevaluated to test whether more efficient implementations or better algorithms are available and can be replaced with substantial speed improvements. This benchmarking is facilitated by standardized options and interfaces and standardized data sets, and we make the data sets we produced here available for reuse. Second, before investing substantial effort in developing nonstandard hardware or approaches to accelerate a specific algorithm, it is worth checking whether a better CPU-based algorithm is available and then, if it is, optimizing that other algorithm instead. Finally, caution is warranted in generalizing which pipeline steps a given algorithm or implementation is used for. Although Minimap2 and Bowtie2 both fundamentally perform the same task (approximate string match to a

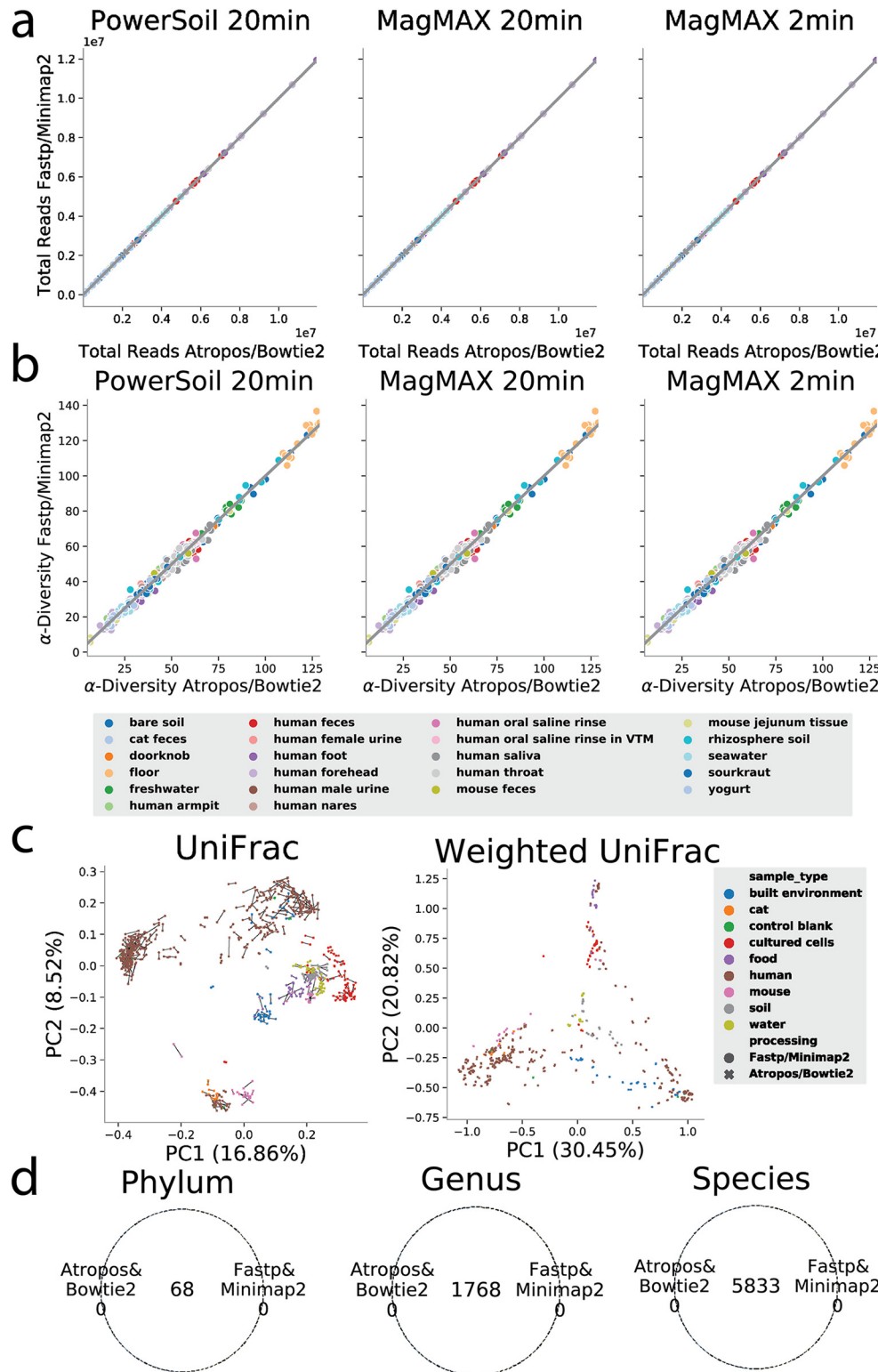

**FIG 2** When comparing broad sets of extraction kits and sample types, Minimap2/Fastp processing results do not differ in biological interpretation compared to current processing methods. (a and b) Comparison of total reads passing the filter (a) and Faith's phylogenetic diversity (b) for Fastp/Minimap2 (y axes) and Atropos/Bowtie2 (x axes) colored by sample type. (c) Principal coordinate analysis (PCoA) on unweighted (left) and weighted (right) UniFrac compared between Fastp/Minimap2 (circles) and Atropos/Bowtie2 (cross) colored by sample source environment. (d) Comparison of shared features between processing methods fastp/Minimap2 and Atropos/Bowtie2 at the phylum, genus, and species taxonomic levels.

database, albeit with different mechanisms), Minimap2's failure on the taxonomic assignment task warrants further investigation to test whether the algorithm could be adapted to this task or whether there are fundamental limitations.

Our current work therefore provides an important practical improvement with a speedup in common metagenomics preprocessing tasks, which we have already made available to the community via incorporation into Qiita (20). Future work will be needed to assess and adapt alignment-free approaches, which often provide improvements in runtime over alignment methods, for both host-filtering and taxonomic assignment tasks. These advancements also point the way toward further optimization that will allow real-time or near-real-time use of metagenomic and/or metatranscriptomic data in clinical decision making, where time is often of the essence.

## SUPPLEMENTAL MATERIAL

Supplemental material is available online only.

**TEXT S1**, DOCX file, 0.01 MB.
**FIG S1**, TIF file, 2.4 MB.
**FIG S2**, TIF file, 2.7 MB.
**TABLE S1**, XLSX file, 0 MB.
**TABLE S2**, XLSX file, 0.02 MB.

## ACKNOWLEDGMENTS

This work was supported in part by CRISP, one of six centers in JUMP, a Semiconductor Research Corporation (SRC) program sponsored by DARPA (https://crisp.engineering.virginia.edu/). J.P.S. was supported by NIH/NIGMS IRACDA K12 GM068524.

We declare that we have no competing interests.

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
