## [Reviewer comments · mSystems]

Swapping metagenomics preprocessing pipeline components offers speed and sensitivity increases

George Armstrong, Cameron Martino, Justin Morris, Behnam Khaleghi, Jaeyoung Kang, Jeff Dereus, Qiyun Zhu, Daniel Roush, Daniel McDonald, Antonio Gonzalez, Justin Shaffer, Carolina Carpenter, Mehrbod Estaki, Stephen Wandro, Sean Eilert, Ameen Akel, Justin Eno, Ken Curewitz, Austin Swafford, Niema Moshiri, Tajana Rosing, and Rob Knight

Corresponding Author(s): Rob Knight, UCSD School of Medicine

Review Timeline:

Submission Date:	November 15, 2021
Editorial Decision:	December 17, 2021
Revision Received:	February 14, 2022
Accepted:	February 18, 2022

Editor: Rachel Mackelprang

Reviewer(s): Disclosure of reviewer identity is with reference to reviewer comments included in decision letter(s). The following individuals involved in review of your submission have agreed to reveal their identity: Zhong Wang (Reviewer #2)

Transaction Report:

DOI: <https://doi.org/10.1128/msystems.01378-21>

December 17, 2021

Dr. Rob Knight
University of California, San Diego
La Jolla

Re: mSystems01378-21 (Swapping metagenomics preprocessing pipeline components offers speed and sensitivity increases)

Dear Dr. Rob Knight:

Thank you for submitting your manuscript to mSystems. We have completed our review and I am pleased to inform you that, in principle, we expect to accept it for publication in mSystems. However, acceptance will not be final until you have adequately addressed the reviewer comments.

Preparing Revision Guidelines

Sincerely,

Rachel Mackelprang

Editor, mSystems

Journals Department
Reviewer comments:

Reviewer #1 (Comments for the Author):

summary

In this study, the authors explored the feasibility of replacing a widely used metagenomics preprocessing stack, atropos+Bowtie2, with fastp+minimap2, to achieve better performance in terms of running time. The motivation was the observation that host contamination filtering for large metagenomics sequencing projects appeared to be a major performance bottleneck.

The inputs of the pipeline(s) are: illumina short reads and a human reference genome. The outputs are: alignment result of primer-trimmed reads. There were three evaluations, on the following datasets, respectively: 1) mixture of simulated short reads with illumina error profile, generated from bacterial assemblies and a human reference genome; 2) mixture of real libraries of 2 metagenomic samples and human libraries of IGSR phase 3 WXS; 3) mixture of real metagenomic libraries from various, known sources, described in a previous study; this dataset offers the opportunity to test against different DNA extraction protocols on the same underlying biosample.

Overall, the study offers a reference for researchers who might be interested in swapping in minimap2 for bowtie2 for this particular preprocessing step (assuming host is human). The manuscript is nicely constructed and written, although it could be improved by adding or revising some details listed below.

major remarks

1. For the read simulation described on the bottom of page 4 (line 87-91), the simulated human reads based on the human reference genome are probably easier to align than most real human reads because they do not have variants. I would like to suggest the authors to perform the simulation using a different reference (e.g. reference assembly built for non-white ethnic groups, or T2T CHM13, or Hifi-based contigs) than the alignment reference, if this section is needed.
2. For the not-simulated experiment at page 5, line 105+, it might be better to use WGS human datasets rather than WXS. Even combining low coverage WGS libraries from IGSR pilot runs might be more suitable than WXS. Alternatively, publicly available illumina WGS from [GIAB](<https://www.nature.com/articles/sdata201625>) or other sources could be good candidates.
3. I wonder if the minimap2 run in figure 1d included indexing time (text: page 5, line 101-102), which could be a significant overhead for the 1M dataset and thus the explanation for the non-constant performance in 1d. However, on my side minimap2 2.18-r1028-dirty uses about 1.5 minutes to process hs38 and write a mmi file (`minimap2 -t12 -x sr -c -d` with /usr/bin/time; 2.1GHz x86_64 CPU`). In figure 1d, it reported a speed of roughly 30k reads per second for 1M reads, which translates to around 33s for the whole run, so very likely it was not building the index from scratch. And per [documentation] (<https://lh3.github.io/minimap2/minimap2.html>), minimap2 by default takes 500M bases into memory in each mini-batch (switch ``-K``). The mini-batching should be a minor influence for the rather small test datasets used in figure 1d. Therefore, the non-constant performance remains unexpected. It would be nice if the authors could double-check this result, or test on larger datasets to effectively ignore small overheads and fluctuations.
4. The manuscript will greatly benefit from a section or a supplementary table describing details of the experiments. While some probably do not have significant impact on the reported results, for the sake of reproducibility and documentation, I think it would be reasonable to include the following information: version number of the tools used, commands used for alignment & simulation & evaluation, commands used for performance evaluation, the frequency of the CPU used.
5. Similar to #4, it would be nice to have the accession IDs or the urls for publicly or restricted access datasets used in the evaluations, namely the soil and mouse fecal metagenomics data in line 108-109 and the datasets from reference#3 (aka. ERP124610 and Qiita 12201). (If data access is actually described in some publisher forms, sorry and please ignore this question. On my side I can only see the contents of: manuscript (text or merged) + figures(4) + tables(2).)

minor remarks

Suggestions or need clarifications:

1. Line 129-134 (Woltka), it is not very clear which dataset did this experiment use (and which mode did bowtie2 use). And it might be helpful to provide a brief observation/explanation as to why minimap2 was less suitable here, as until this part, most alignment benchmarks have been focused on human reads and reference. Abstract implied that this was due to "bowtie2's (higher) specificity", but some elaboration or citation would make the writing easier to follow. (see also major#5 about documenting command lines.)
2. Figure S1's x-axis label, typo: minimap2.
3. Since the motivation of this study is to seek a performant workflow (page 4, line 71-73), have the authors tried any alignment-free approach, or hybrid (e.g. pre-filter with kmer-based tool, align only the ambiguous reads)? If so, how were they?

It's up to the authors whether to respond to any of the following comments:

1. For hs38 as the alignment reference, `GCF_000001405.39` has alt contigs. Minimap2 is not alt-aware without a list of alt contig names (see manpage and [this](https://github.com/lh3/minimap2/issues/72)). `GCA_000001405.15` might be a better choice.
2. Looking at figure 1c-left (UniFrac) and the main text, I guess this and the figure 1c-right (Weighted UniFrac) were meant to show that points representing different stacks located together in the feature space. It is kind of hard to see though, as points sit on top of each other. I guess it would be clearer if the points are smaller in diameter, and the figures exported in vector format.

Reviewer #2 (Comments for the Author):

Armstrong et al. developed their own datasets to validate the performance improvement claimed by the more recent software tools (Fastp and Minimap2) over older ones (atropos and bowtie2). They found Fastp and MiniMap2 can fully replace atropos and bowtie2 in adapter sequence filtering and contaminant screening, respectively. Switching to Fastp and Minimap2 offered the benefit in computing efficiency. However, Minimap2 did not replace bowtie2 in taxonomy prediction. The manuscript is well written and easy to follow.

My major concern is there are no descriptions of what parameters were used to carry out the above tests. The authors may have omitted the method section? Specifically, minimap2 was designed for long-read mapping by aligning multiple short minimizers, but with short reads, its accuracy may be limited, especially when the reference set used for taxonomy contains multiple related genomes. Exploring different sets of parameters such as larger minimizers may provide more insights. The authors explored different parameter sets of bowtie2, and it would be fair to do the same for minimap2.

Minor comments:

1. When comparing computing efficiency, it would be nice to compare peak memory usage as well, since a longer computing time can be due to a trade-off in space efficiency.
2. Supplemental Figure S2a only showed the false discovery rate of minimap2, for some reason those of bowtie2 were not shown. Please explain.

Reviewer #3 (Comments for the Author):

This document describes a comparison between the Fastp/Minimap2 and Atropos/Bowtie2 tools in the context of detecting populations present in a metagenomic sample. The paper uses this comparison to say that the use of tools and algorithms based on technologies that do not use classical CPUs (GPU, FPGA), is not necessary.

Although the comparison methods and datasets used seem to me to be quite relevant, which makes these comparisons useful for the readers. There are a number of points I have problems with:

My comments are sorted in order of importance:

- I think it is necessary to include more tools in the benchmark. It would make it more relevant and useful for the readers. I was surprised not to see at least a comparison with bwa-mem (or bwa-mem2), which I think is between bowtie2 and minimap2 in terms of computation time.
- The fact that minimap2 is faster than bowtie2 does not seem to me to be a surprise or a recent discovery in the bioinformatics community. And even if I agree that we don't need (in a lot of cases) to use GPU or FPGA for these alignment tasks, I find the arguments presented in this paper quite weak. Especially since the paper doesn't compare minimap2 and bowtie2 to this kind of tools, or give at least an estimation of the computation time.
- The fastp and Atropos tools are only compared individually on their computation time, the quality of their results is only compared in the whole pipeline. I would have appreciated more details on the results of these two tools.

summary

In this study, the authors explored the feasibility of replacing a widely used metagenomics preprocessing stack, atropos+Bowtie2, with fastp+minimap2, to achieve better performance in terms of running time. The motivation was the observation that host contamination filtering for large metagenomics sequencing projects appeared to be a major performance bottleneck.

The inputs of the pipeline(s) are: illumina short reads and a human reference genome. The outputs are: alignment result of primer-trimmed reads. There were three evaluations, on the following datasets, respectively: 1) mixture of simulated short reads with illumina error profile, generated from bacterial assemblies and a human reference genome; 2) mixture of real libraries of 2 metagenomic samples and human libraries of IGSR phase 3 WXS; 3) mixture of real metagenomic libraries from various, known sources, described in a previous study; this dataset offers the opportunity to test against different DNA extraction protocols on the same underlying biosample.

Overall, the study offers a reference for researchers who might be interested in swapping in minimap2 for bowtie2 for this particular preprocessing step (assuming host is human). The manuscript is nicely constructed and written, although it could be improved by adding or revising some details listed below.

major remarks

1. For the read simulation described on the bottom of page 4 (line 87-91), the simulated human reads based on the human reference genome are probably easier to align than most real human reads because they do not have variants. I would like to suggest the authors to perform the simulation using a different reference (e.g. reference assembly built for non-white ethnic groups, or T2T CHM13, or Hifi-based contigs) than the alignment reference, if this section is needed.
2. For the not-simulated experiment at page 5, line 105+, it might be better to use WGS human datasets rather than WXS. Even combining low coverage WGS libraries from IGSR pilot runs might be more suitable than WXS. Alternatively, publicly available illumina WGS from [GIAB](<https://www.nature.com/articles/sdata201625>) or other sources could be good candidates.
3. I wonder if the minimap2 run in figure 1d included indexing time (text: page 5, line 101-102), which could be a significant overhead for the 1M dataset and thus the explanation for the non-constant

performance in 1d. However, on my side minimap2 2.18-r1028-dirty uses about 1.5 minutes to process hs38 and write a mmi file (`minimap2 -t12 -x sr -c -d` with /usr/bin/time; 2.1GHz x86_64 CPU`). In figure 1d, it reported a speed of roughly 30k reads per second for 1M reads, which translates to around 33s for the whole run, so very likely it was not building the index from scratch. And per [documentation](<https://lh3.github.io/minimap2/minimap2.html>), minimap2 by default takes 500M bases into memory in each mini-batch (switch `-K``). The mini-batching should be a minor influence for the rather small test datasets used in figure 1d.

Therefore, the non-constant performance remains unexpected. It would be nice if the authors could double-check this result, or test on larger datasets to effectively ignore small overheads and fluctuations.

4. The manuscript will greatly benefit from a section or a supplementary table describing details of the experiments. While some probably do not have significant impact on the reported results, for the sake of reproducibility and documentation, I think it would be reasonable to include the following information: version number of the tools used, commands used for alignment & simulation & evaluation, commands used for performance evaluation, the frequency of the CPU used.

5. Similar to #4, it would be nice to have the accession IDs or the urls for publicly or restricted access datasets used in the evaluations, namely the soil and mouse fecal metagenomics data in line 108-109 and the datasets from reference#3 (aka. ERP124610 and Qiita 12201). (If data access is actually described in some publisher forms, sorry and please ignore this question. On my side I can only see the contents of: manuscript (text or merged) + figures(4) + tables(2).)

minor remarks

Suggestions or need clarifications:

1. Line 129-134 (Woltka), it is not very clear which dataset did this experiment use (and which mode did bowtie2 use). And it might be helpful to provide a brief observation/explanation as to why minimap2 was less suitable here, as until this part, most alignment benchmarks have been focused on human reads and reference. Abstract implied that this was due to "bowtie2's (higher) specificity", but some elaboration or citation would make the writing easier to follow. (see also major#5 about documenting command lines.)

2. Figure S1's x-axis label, typo: minimap2.

3. Since the motivation of this study is to seek a performant workflow (page 4, line 71-73), have the authors tried any alignment-free approach, or hybrid (e.g. pre-filter with kmer-based tool, align only the ambiguous reads)? If so, how were they?

It's up to the authors whether to respond to any of the following comments:

1. For hs38 as the alignment reference, `GCF_000001405.39` has alt contigs. Minimap2 is not alt-aware without a list of alt contig names (see manpage and [this](https://github.com/lh3/minimap2/issues/72)). `GCA_000001405.15` might be a better choice.

2. Looking at figure 1c-left (UniFrac) and the main text, I guess this and the figure 1c-right (Weighted UniFrac) were meant to show that points representing different stacks located together in the feature space. It is kind of hard to see though, as points sit on top of each other. I guess it would be clearer if the points are smaller in diameter, and the figures exported in vector format.

Dear Rachel,

Thank you for the time you have spent handling this manuscript, and thanks also to the reviewers for their insightful and helpful feedback. We have responded below point by point, and hope that the revised manuscript adequately addresses all the concerns.

Best,

Rob

Reviewer #1 (Comments for the Author):

summary

In this study, the authors explored the feasibility of replacing a widely used metagenomics preprocessing stack, atropos+Bowtie2, with fastp+minimap2, to achieve better performance in terms of running time. The motivation was the observation that host contamination filtering for large metagenomics sequencing projects appeared to be a major performance bottleneck.

The inputs of the pipeline(s) are: illumina short reads and a human reference genome. The outputs are: alignment result of primer-trimmed reads. There were three evaluations, on the following datasets, respectively: 1) mixture of simulated short reads with illumina error profile, generated from bacterial assemblies and a human reference genome; 2) mixture of real libraries of 2 metagenomic samples and human libraries of IGSR phase 3 WXS; 3) mixture of real metagenomic libraries from various, known sources, described in a previous study; this dataset offers the opportunity to test against different DNA extraction protocols on the same underlying biosample.

Overall, the study offers a reference for researchers who might be interested in swapping in minimap2 for bowtie2 for this particular preprocessing step (assuming host is human). The manuscript is nicely constructed and written, although it could be improved by adding or revising some details listed below.

major remarks

1. For the read simulation described on the bottom of page 4 (line 87-91), the simulated human reads based on the human reference genome are probably easier to align than most real human reads because they do not have variants. I would like to suggest the authors to perform the simulation using a different reference (e.g. reference assembly built for non-white ethnic groups, or T2T CHM13, or Hifi-based contigs) than the alignment reference, if this section is needed.

We thank the reviewer for this advice. We have now aligned the simulation data to a different reference from which they were generated (T2T CHM13). We have updated Figure 1a-d to reflect this and note it in the text on line 94.

2. For the not-simulated experiment at page 5, line 105+, it might be better to use WGS human datasets rather than WXS. Even combining low coverage WGS libraries from IGSR pilot runs might be more suitable than WXS. Alternatively, publicly available illumina WGS from [GIAB](<https://www.nature.com/articles/sdata201625>) or other sources could be good candidates.

We thank the reviewer for the concern, and we agree that choice of data for the mixtures is important. For WGS it is very possible that there are likely non-negligible amounts of microbial reads (PMID: 32214244), and therefore we have used WES for the mock mixture experiments. We have added additional text to clarify this point on lines 114-115.

3. I wonder if the minimap2 run in figure 1d included indexing time (text: page 5, line 101-102), which could be a significant overhead for the 1M dataset and thus the explanation for the non-constant performance in 1d. However, on my side minimap2 2.18-r1028-dirty uses about 1.5 minutes to process hs38 and write a mmi file (`minimap2 -t12 -x sr -c -d` with /usr/bin/time; 2.1GHz x86_64 CPU`). In figure 1d, it reported a speed of roughly 30k reads per second for 1M reads, which translates to around 33s for the whole run, so very likely it was not building the index from scratch. And per [documentation](<https://lh3.github.io/minimap2/minimap2.html>), minimap2 by default takes 500M bases into memory in each mini-batch (switch ``-K``). The mini-batching should be a minor influence for the rather small test datasets used in figure 1d. Therefore, the non-constant performance remains unexpected. It would be nice if the authors could double-check this result, or test on larger datasets to effectively ignore small overheads and fluctuations.

We thank the reviewer for this concern. The times do not include indexing, as observed. In order to address the effect of overheads, we benchmarked Minimap2 on a larger number of reads until the performance remained constant. This has been updated in Figure 1d, and we updated the text on lines 103-104 to reflect that this non-constant performance is not expected after the number of reads continues to increase.

4. The manuscript will greatly benefit from a section or a supplementary table describing details of the experiments. While some probably do not have significant impact on the reported results, for the sake of reproducibility and documentation, I think it would be reasonable to include the following information: version number of the tools used, commands used for alignment & simulation & evaluation, commands used for performance evaluation, the frequency of the CPU used.

We thank the reviewer for this request. We have added a supplemental materials section that includes the requested details.

5. Similar to #4, it would be nice to have the accession IDs or the urls for publicly or restricted access datasets used in the evaluations, namely the soil and mouse fecal metagenomics data in line 108-109 and the datasets from reference#3 (aka. ERP124610 and Qiita 12201). (If data access is actually described in some publisher forms, sorry and please ignore this question. On my side I can only see the contents of: manuscript (text or merged) + figures(4) + tables(2).)

We have also added this information to the supplementary materials.

minor remarks

Suggestions or need clarifications:

1. Line 129-134 (Woltka), it is not very clear which dataset did this experiment use (and which mode did bowtie2 use). And it might be helpful to provide a brief observation/explanation as to why minimap2 was less suitable here, as until this part, most alignment benchmarks have been focused on human reads and reference. Abstract implied that this was due to "bowtie2's (higher) specificity", but some elaboration or citation would make the writing easier to follow. (see also major#5 about documenting command lines.)

We thank the reviewer for making this clarification. We have added dataset/experiment details to the supplemental materials. We added an additional clarification on the explanation of woltka results on lines 140-144.

2. Figure S1's x-axis label, typo: minimap2.

We thank the reviewer for this correction. We have updated the label.

3. Since the motivation of this study is to seek a performant workflow (page 4, line 71-73), have the authors tried any alignment-free approach, or hybrid (e.g. pre-filter with kmer-based tool, align only the ambiguous reads)? If so, how were they?

We thank the reviewer for this comment. We did not benchmark alignment-free methods, as in this work we focused only on alignment-based methods. Future work will be needed to assess and adapt alignment-free approaches comprehensively. We have added this statement into the manuscript on lines 163-165.

It's up to the authors whether to respond to any of the following comments:

1. For hs38 as the alignment reference, `GCF_000001405.39` has alt contigs. Minimap2 is not alt-aware without a list of alt contig names (see manpage and [this](<https://github.com/lh3/minimap2/issues/72>)). `GCA_000001405.15` might be a better choice.

We thank the reviewer for making note of this. We have updated the reference genome to GCA_009914755.3 for the simulation data per major comment 1.

2. Looking at figure 1c-left (UniFrac) and the main text, I guess this and the figure 1c-right (Weighted UniFrac) were meant to show that points representing different stacks located together in the feature space. It is kind of hard to see though, as points sit on top of each other. I guess it would be clearer if the points are smaller in diameter, and the figures exported in vector format.

We appreciate the feedback on the figure. We have updated Figure 2c with smaller points to help reveal the pattern, and note that we have uploaded the figure at the maximal resolution allowed by the submission portal.

Reviewer #2 (Comments for the Author):

Armstrong et al. developed their own datasets to validate the performance improvement claimed by the more recent software tools (Fastp and Minimap2) over older ones ((atropos and bowtie2). They found Fastp and MiniMap2 can fully replace astropos and bowtie2 in adapter sequence filtering and contaminant screening, respectively. Switching to Fastp and Minimap2 offered the benefit in computing efficiency. However, Minimap2 did not replace bowtie2 in taxonomy prediction. The manuscript is well written and easy to follow.

My major concern is there are no descriptions of what parameters were used to carry out the above tests. The authors may have omitted the method section? Specifically, minimap2 was designed for long-read mapping by aligning multiple short minimizers, but with short reads, its accuracy may be limited, especially when the reference set used for taxonomy contains multiple related genomes. Exploring different sets of parameters such as larger minimizers may provide more insights. The authors explored different parameter sets of bowtie2, and it would be fair to do the same for minimap2.

Thank you for the comment. We have included a supplementary section that includes all of these details. For minimap2, although it was designed for long reads, we specifically used the short-read preset parameters for minimap2—this provides improvements over the long-read optimized version, and we have now mentioned this decision to use the short-read preset on lines 92-93.

Minor comments:

1. When comparing computing efficiency, it would be nice to compare peak memory usage as well, since a longer computing time can be due to a trade-off in space efficiency.

We agree that it is helpful to include memory in these benchmarks. We have added a figure panel (Fig. S1C) to show these results and made mention of the memory benchmarks on lines 109-110.

2. Supplemental Figure S2a only showed the false discovery rate of minimap2, for some reason those of bowtie2 were not shown. Please explain.

We thank the reviewer for the comment. We have added a note to the legend on lines 273-274 that a bar that was not visible indicates a value of zero.

Reviewer #3 (Comments for the Author):

This document describes a comparison between the Fastp/Minimap2 and Atropos/Bowtie2 tools in the context of detecting populations present in a metagenomic sample. The paper uses this comparison to say that the use of tools and algorithms based on technologies that do not use classical CPUs (GPU, FPGA), is not necessary.

Although the comparison methods and datasets used seem to me to be quite relevant, which makes these comparisons useful for the readers. There are a number of points I have problems with:

My comments are sorted in order of importance:

- I think it is necessary to include more tools in the benchmark. It would make it more relevant and useful for the readers. I was surprised not to see at least a comparison with bwa-mem (or bwa-mem2), which I think is between bowtie2 and minimap2 in terms of computation time.

We thank the reviewer for the suggestion to benchmark an additional aligner. We have added BWA to the simulation benchmarks (Fig. 1), mention on line 98 that it is an order of magnitude slower than minimap2, and note on lines 106-107 that we have only considered minimap2 and bowtie2 for the end-to-end benchmarks since the goal is to compare the results of the status quo pipeline to the new fastest method. We note that although we had planned to run BWA on the real data as well, it is so slow that these runs did not complete before the resubmission deadline.

- The fact that minimap2 is faster than bowtie2 does not seem to me to be a surprise or a recent discovery in the bioinformatics community. And even if I agree that we don't need (in a lot of cases) to use GPU or FPGA for these alignment tasks, I find the arguments presented in this paper quite weak. Especially since the paper doesn't compare minimap2 and bowtie2 to this kind of tools, or give at least an estimation of the computation time.

We thank the reviewer for their comment. We have modified the text on lines 45-46 and lines 79-80 to emphasize that the speed results are only half of the story—demonstrating the biological similarity of the results is also a key component of the analysis.

Furthermore, we have updated the wording on line 75-76 to indicate that the key result was to produce benchmarks of candidate methods that could be further accelerated by GPU or FPGA, rather than indicating that CPU methods will be sufficient for the needs of all users.

- The fastp and Atropos tools are only compared individually on their computation time, the quality of their results is only compared in the whole pipeline. I would have appreciated more details on the results of these two tools.

We have added a figure panel (Fig. S1A) that compares the results of fastp and atropos individually and note that their outputs are similar on lines 104-106.

February 18, 2022

Prof. Rob Knight
UCSD School of Medicine
9500 Gilman Drive
MC 0602
La Jolla, CA 92093

Re: mSystems01378-21R1 (Swapping metagenomics preprocessing pipeline components offers speed and sensitivity increases)

Dear Prof. Rob Knight:

Your manuscript has been accepted, and I am forwarding it to the ASM Journals Department for publication. For your reference, ASM Journals' address is given below. Before it can be scheduled for publication, your manuscript will be checked by the mSystems production staff to make sure that all elements meet the technical requirements for publication. They will contact you if anything needs to be revised before copyediting and production can begin. Otherwise, you will be notified when your proofs are ready to be viewed.

Publication Fees:

We recognize that the video files can become quite large, and so to avoid quality loss ASM suggests sending the video file via <https://www.wetransfer.com/>. When you have a final version of the video and the still ready to share, please send it to mSystems staff at mssystemsjournal@msubmit.net.

For mSystems research articles, if you would like to submit an image for consideration as the Featured Image for an issue, please contact mSystems staff at mssystemsjournal@msubmit.net.

Sincerely,

Rachel Mackelprang
Editor, mSystems

Journals Department
Supplemental Material: Accept
Supplemental Material: Accept
Table S1: Accept
Table S2: Accept
Supplemental Material: Accept